# Micron- and Nanosized Alloy Particles Made by Electric Explosion of W/Cu-Zn and W/Cu/Ni-Cr Intertwined Wires for 3D Extrusion Feedstock

**DOI:** 10.3390/ma16030955

**Published:** 2023-01-19

**Authors:** Marat Lerner, Konstantin Suliz, Aleksandr Pervikov, Sergei Tarasov

**Affiliations:** 1Faculty of Physics and Technology, Tomsk State University, 634050 Tomsk, Russia; 2Institute of Strength Physics and Material Science, Siberian Branch of Russian Academy of Science, 634055 Tomsk, Russia

**Keywords:** W/Cu composite, powders, exploding wires, nanoparticles

## Abstract

A novel approach to electric explosion of intertwined wires to obtain homogeneous powder mixtures intended for preparing feedstock for extrusion 3D printing has been applied. The powder were composed of spherical micron- and nano-sized W/Cu particles in-situ alloyed by Zn and Ni during electric explosion of intertwined dissimilar metal wires is offered. The mean particle size measured by micron-sized particles was not more than 20 μm. The average number size of these particles was 3 μm and it was dependent on the energy input. The powders contained phases such as α-W, β-W/W_3_O as well as FCC α-Cu(Zn) and α-Cu(Ni) solid solutions with the crystalline lattice parameters 3.629 and 3.61 A, respectively.

## 1. Introduction

The W-Cu pseudoalloys are widely used in different industries as functional materials possessing high electric and heat conductances, and high resistance to erosion, wear and radiation [1,2,3]. The use of casting for producing the W/Cu pseudoalloys is limited due to great difference in their thermodynamic and physical parameters, and therefore powder metallurgy is the most promising method that allows improving both the W/Cu pseudoalloy’s physico-mechanical and functional characteristics [1].

The W/Cu pseudoalloys may be produced also using mechanical alloying [4] as well as chemical [5,6] and physical [7,8] methods. An alternative method is used such as electric explosion of wires that allows obtaining completely spherical particles [9] by passing a current pulse with a current density as high as 10^6^–10^9^ A/cm^2^ through a metallic wire [10]. The instant heating changes the wire metal state from the solid to a two-phased state defined as explosion products composed of micron- and nano-sized droplets + low ionized plasma [11]. The next stage will be cooling and condensation of the expanding products into spherical micron- and nanosized particles [12,13]. An advantage of this method would be an opportunity for particle size adjustment by varying the energy input into a wire and thus varying both the size and volume fraction of the micro- and nano-sized components in the mixture [9,14].

It was shown [15,16] that the use of homogeneous powder blends prepared from both micron- and nanosized particles allows obtaining feedstocks with improved characteristics required for fabricating materials with high physico-mechanical properties using either extrusion 3D printing or metal injection molding (MIM).

The use of simultaneous electric explosion of two and more dissimilar metal wires allows obtaining particles composed of two or more elements [17,18]. Varying the wire diameter and keeping its length constant makes it possible to vary the element content in a particle as well as to modify its macrostructure for obtaining either Janus-like or core-shell particles [19,20].

The as-synthesized W/Cu particle metal may oxidize when exposed to air [21] and it is especially crucial for the copper whose electric resistance greatly depends on the content of oxygen. Therefore, particles should be subjected to hydrogen reduction before using any consolidation method. Development of processes that permit either full or partial avoidance of the copper particle oxidation is urgent for preparation of the copper-base feedstock for both extrusion 3D printing and MIM. Depending on the chemical composition of a polymer binder, all technological operations intended for homogenization of the feedstocks or those performed in 3D extrusion printing are carried out at the temperatures in the range 100–250 °C, i.e., temperatures high enough for oxidizing the copper nanoparticles [22].

An alternative solution to this problem may be alloying the copper component by such metals as Zn, Ni, Sn, Ag and thus reducing its oxidation degree, for example, in fabricating the water-base copper-containing conductive ink for printing the integrated electric circuits [23].

An analogous approach has been used in this work for producing micron- and nanosized W/Cu-Zn and W/Cu/Ni-Cr particles by electric explosion of wires in argon. The in situ alloying of W/Cu particles with the above indicated elements would allow improving mechanical and physico-mechanical characteristics of the feedstock and consolidated components [1]. Alloying the particles with elements such as Zn, Ni and Cr would improve wetting between W and Cu components, intensify sintering (Ni) and increase their hardness due to precipitation of Cr in the copper matrix [24].

A part of the work must be devoted to analyzing the energy input effect on both particle size distribution and formation of phases. Establishing these relationships is necessary for producing the quality feedstock powders with different particle size distributions to be used for tailoring the characteristics of either consolidated or additively manufactured composites.

## 2. Experimental Procedure

### 2.1. Electric Explosion Synthesis of Multimetallic Particles

The electric explosion of intertwined dissimilar metal W/Cu-37%Zn and W/Cu/79%Ni-21%Cr wires was used to obtain the W/Cu-Zn and W/Cu/Ni/Cr powders composed of both micro- and nano-sized particles in an electric explosion machine whose functioning utilized the RLC circuit principle [25]. The explosion frequency was 0.5 Hz and the explosion parameters were determined from the current dependencies *I(t)* measured using the Rogowski belt. The moment of time corresponding to the explosion of wire was determined by detecting a local minimum at the *dI/dt* curve [26]. The amount of energy *E* input into the wire was calculated according to Equation (1) [27] as follows:(1)$$E(t)=U_{0}\int_{0}^{t}I(t)dt-\frac{1}{2C}{\left[\int_{0}^{t}I(t)dt\right]}^{2}-\frac{LI^{2}(t)}{2}-R\int_{0}^{t}I^{2}(t)dt$$
where *U_0_* is the capacitor bank charging voltage (kV), *C* is the total electric capacity of the bank (μF), *L* is the inductance of the *RLC*-circuit (~0.75 μH), *R* is the ohmic resistance of the *RLC* circuit (~0.086 Ohm) and Σ*E_s_* is the total sublimation energy of the wires intertwined calculated from the reference data [28]. All the process parameters used for producing both W/Cu/Zn and W/Cu/Ni/Cr powders are shown in Table 1. The wire geometry parameters provided the relative contents of the constituent metals as follows (mass %): W-17Cu-9.4Zn, W-23Cu-4.4Ni-1.1Cr. The energy input was varied by varying the magnitude of the charging voltage *U_0_*.

### 2.2. Characterization of Powders

Two powder lots of 300 g each were produced and then passivated in air for 24 h to exclude spontaneous oxidizing in further handling them. The basic diagram of electric explosion of wire process and machine used for producing the W/Cu/Zn and W/Cu/Ni/Cr powders is shown in Figure 1a together with the macrograph of intertwined wires (Figure 1b). The intertwined wire pitch was 1 wind per 1 cm of length. It was found out that this value allows neglecting the wire deformation effect on the energy release in passing the current through it [29].

The powder production process is as follows. The intertwined wires are reeled up and fixed inside the wire feeding device (Figure 1a, pos.1). The high-voltage electrode is connected with the energy capacitor (Figure 1a, pos.8) via an air-gap discharge device and then displaced to set the gap between the high-voltage and grounded electrodes inside the explosion chamber (Figure 1a, pos.2). This gap defines the length of wire to be exploded. On such an adjustment of the inter-electrode gap, the chamber is closed and evacuated until reaching the residual air pressure of ~1 Pa. The next stage is filling the chamber with argon up to its residual pressure of 0.3 MPa. On reaching this pressure level, the capacitor system (Figure 1a, pos.8) starts charging as well as both argon circulation system (Figure 1a, pos.7) and wire feed device (Figure 1a, pos.1) are activated

Adjusting the angular velocity of the wire feeder rollers allows specifying the required value of the explosion frequency. The electric explosion of wire occurs when this wire touches the high-voltage electrode. The explosion products are carried away by the argon flow from the explosion chamber (Figure 1a, pos.2) to the separator (Figure 1a, pos.3) where the non-exploded wire chunks as well as large >50 μm in size drops are separated from the powders which then carried away to the cyclone apparatus (Figure 1a, pos.5) and then settle in the bin (Figure 1a, pos.6). On producing the desired amount of powder, the argon pressure is decreased to the atmospheric level and powder is passivated in air.

The particle micrographs were obtained using an SEM instrument, the Quanta 200 3D (FEI Company, Hillsboro, OR, USA) attached with an EDS analyzer. The particle size distribution was reconstructed from measuring the particle sizes in those SEM micrographs. An X-ray diffractometer, the Shimadzu XRD 6000, CuKa (Shimadzu, Kyoto, Japan), was used to obtain the powder diffractograms whose further identification was carried out using the PDF2 database and XPowder 2004 software The coherent scattering area sizes (*d_csr_*) as well as crystalline lattice microdistortions were determined according to the Williams-Hall method [30].

## 3. Results and Discussion

It is known [9] that the particle size distribution resulting from an electric explosion of wire (EEW) is determined by the energy input with respect to total wire sublimation energy, i.e., *E*/Σ*E_s_* so that varying this ratio it becomes possible to produce either micron- or nano-sized particles. If *E*/Σ*E_s_* value is within the 0.3-0.9 range then the major part of the powder is represented with both micron- and submicron-sized particles. The nanosized particles are the main fraction at *E*/Σ*E_s_* > 1.5. When *E*/Σ*E_s_* values are within the 0.9–1.5 range the powder is a homogeneous mixture composed of both micron- and nanosized particles.

Apart from the RLC-circuit parameters, the *E*/Σ*E_s_* ratio magnitude is also dependent on the wire material, for instance, it is possible to provide an energy input much higher than the total sublimation energy using wires made of metals possessing high electric conductance and simultaneously low-melting points [11]. This kind of electric explosion may be used for producing mainly nanosized particles of metals or alloys. According to [31], the instant heating of a brass wire by passing through it a current pulse resulted in its full sublimation and then condensation of the gaseous phase into the nanosized particles. In comparison with the wire composition, the zinc-lean *a*-Cu solid solution and zinc-rich Cu_5_Zn_8_ phases are formed under conditions like that. The evaporation of the brass wire creates conditions for enriching the nanoparticles with zinc because of considerable differences in melting and boiling among the W, Cu and Zn metals. This enriching leads to formation of ZnO on the particle surfaces, which is detrimental for the W/Cu/Zn powder consolidation.

It should be noted, however, that the feasibility of full evaporation of a metal by passing a current pulse through it is still in dispute [26,32].

Electric explosion of wires made of metals characterized by both a low electric conductance (79NiCr21 alloys) and high melting point can be carried out with an energy input comparable to the total sublimation energy [11]; therefore, only micron-sized particles would be obtained [33]. The above-discussed data allow suggesting that alloying the copper particles with refractory elements in the electric explosion conditions can be limited by not suitable particle size distribution of powders obtained from the intertwined W/Cu and 79Ni21Cr wires.

Producing the W/Cu/Zn and W/Cu/Ni/Cr powders composed of both micro- and nanosized particles and containing the desired *a*-W+ *a*-Cu(Zn, Ni) phases is feasible under condition that there is a dependence between the energy input and phases formed

### 3.1. Electric Explosion Synthesis of W/Cu-Zn Powders

The time-dependent oscillograms of current (Figure 1a) and energy (Figure 1b) input recorded during electric explosion of intertwined tungsten and brass wires showed that increasing the charging voltage from 17 to 27 kV resulted in obtaining powders with the energy input ratio *E*/Σ*E_s_* varying within the range 0.5 to 1.25.

The current vs. time dependencies give evidence on existing different electric explosion regimes depending upon the *E*/Σ*E_s-_*ratio. For instance, the explosion regime close to that of short circuiting occurs at *E*/Σ*E_s_* ≈ 0.5. This regime is characterized by the low energy release in the bulk of the metal [34]. There is a local minimum on the *dI/dt* curve at *t* ≈ 2.15 μs that indicates on the sharp change in conductance caused by the explosion of one of the wires. It is obvious that the brass wire would be first to explode in passing the current pulse because of its lower melting point and higher conductance in comparison to the partner wire.

The intertwined wires are connected in parallel circuit so that their equivalent resistance R_wires_ is determined by the resistance R_w_ of the tungsten one, which is increased in passing the current in the time interval 2.15–4.0 μs and overpass the 2(LC)^−0.5^ value at *t* ≈ 4.0 μs. An aperiodic discharge regime is established at *t* > 4.0 μs when partial wire metal is dispersed into melted drops whose diameter is close to that of the wire [35].

More discharge regime changes occur at *E*/Σ*E_s_* ≈ 0.8 and *E*/Σ*E_s_* ≈ 1.25 when two more local minimums appear at the *dI/dt* curves at *t* ≈ 1.9 and *t* ≈ 1.65 μs (marked by crosses in Figure 2b). These minimums can be related to the conductance changes caused by the exploding brass wire. It is only a little of time when the discharge go into oscillation regime which is characterized also as arcing. This transition to the oscillating arc discharge means that R_wires_ < 2(LC)^−0.5^ is true, i.e., there is sharp R_wires_ fall that plausibly is due to discharge occurring either by the brass wire explosion product or by the tungsten wire surface. These discharges may also occur simultaneously as observed by the example of exploding both copper and tungsten wires [36]. It is obvious that if occurred these discharges would interfere with the efficiency of energy input.

The BSE SEM micrographs and corresponding EDS maps of the W/Cu-Zn powders showed they were composed of micron-sized tungsten particles and nano-sized copper alloy ones (Figure 3). The mean particle size is decreased when increasing the *E*/Σ*E_s_* ratio thus making more even the particle distribution histogram. It is noticed that more tungsten particles form at higher *E*/Σ*E_s_* (Figure 3c,f,i). It is suggested that reducing the W content when decreasing the *E*/Σ*E_s_* can be explained by preferential settling of the large > 50μm –sized tungsten particles in the separator bin (Figure 1, pos.4), and vice versa smaller tungsten particles are formed at higher *E*/Σ*E_s_* values whose settling in the separator is less probable.

Typical micrographs of the W/Cu-Zn particles produced with *E*/Σ*E_s_* ≈ 0.5, *E*/Σ*E_s_* ≈ 0.8, *E*/Σ*E_s_* ≈ 1.25 are presented in Figure 4a–c together with the corresponding particle size distributions, which show that the majority of the micron-sized particles were smaller than 10 μm. The increase in *U_0_* from 17 to 27 kV was accompanied by the increase in the *E*/Σ*E_s_* ratio so that the micron-sized particles changed their mean size from 3.8 to 2.8 μm. Note that the increase in *E*/Σ*E_s_* from 0.8 to 1.25 kV has almost no effect on the mean size of the micron-sized particles. Such a situation can be explained by formation a conductive channel at *U_0_* > 22 kV (Figure 2a) that interfered with the energy input and, correspondingly, with the particle size refining under conditions of the experiment.

Figure 5 shows the presence of phases such as α-W, β-W/W_3_O, and FCC α-Cu(Zn) solid solution irrespective of the energy input level. A semi-quantitative analysis of the corresponding XRD peak heights allows us to suggest that the amount of the β-W/W_3_O phase increased with the energy input. Such a finding might have resulted from increasing the cooling rate of the smaller liquid W drops produced by the explosion at higher energy input, and better stabilization of the metastable β-W. The increased amount of the W_3_O may be related to increasing the specific particle surface area, and, consequently more intensive oxidizing.

The FCC α-Cu(Zn) phase with the crystalline lattice parameter a = 3.629 A corresponds to a solid solution of Zn in Cu that also follows from the almost identical distributions of Cu and Zn in the EDS maps (Figure 3).

Table 2 contains data on the size of coherent scattering areas d_csr_ and crystalline lattice microdistortions Δd/d of phases identified in the sample using the XRD. The magnitudes of both d_csr_ and Δd/d decrease with increasing the *E/ΣE_s_* that can be provided by reducing the mean particles size of particles and forming more perfect crystalline lattice, respectively. The latter can be achieved by due to transition to the arc discharge and corresponding reducing the particle cooling rate.

### 3.2. Electric Explosion Synthesis of W/Cu/Ni-Cr Powders

The current and energy input diagrams were obtained during electric explosion of intertwined W/Cu/79%Ni/21%Cr wires at the energy input levels 18, 22 and 27 J (Figure 6a,b). The corresponding *E*/Σ*E_s_* ratio increased from 0.5 to 1.1.

The time dependencies of the current allows us to suggest the existence of different explosion regimes, for instance, the regime close to that of short circuiting occurs for *E*/Σ*E_s_* ≈ 0.5 and *E*/Σ*E_s_* ≈ 0.7. The *dI/dt* curves allow identifying local minimums at *t* ≈ 2.4 and *t* ≈ 2.0 μs which can be related to explosion of the copper wire. The aperiodic regime is established for *E*/Σ*E_s_* ≈ 0.5 and *E*/Σ*E_s_* ≈ 0.7 at *t* > 4.5 μs and *t* > 3.5 μs, respectively.

No explosion regime changes occur for *E*/Σ*E_s_* ≈ 1.1, and *dI/dt* minimum at *t* ≈ 1.7 μs testify on the conductance changes appeared dute to explosion of the copper wire. The next stage is the transition to the oscillating current regime denoted by cross marks in Figure 6a.

The SEM micrographs and EDS element distribution maps (Figure 7) were obtained from the W/Cu/Ni-Cr powder and suggested it was composed of both micron- and nano-sized particles. The micron-sized ones are represented by nickel-chromium and tungsten ones. It was noted above that increasing the *E*/Σ*E_s_* ratio improves the particle size distribution homogeneity due to reduction of the mean particle size. Also the content of W in the W/Cu/Ni-Cr powder is increased analogously to that in the above-discussed W/Cu/Zn powders (Figure 7c,f,i).

The W/Cu/Ni-Cr powder micrographs (Figure 8a–c) and corresponding particle size distributions show that almost all the micron-sized particles were smaller than 10 μm irrespective of the energy input *U_0_*. Their mean size decreased from 4.4 to 3.6 μm in accordance to corresponding increasing of the *E*/Σ*E_s_* ratio. Again, increasing the *E*/Σ*E_s_* from 0.7 to 1.1 had not any essential effect on the micron-sized particles because of formation of the conductive channel at *E*/Σ*E_s_* > 0.7 (Figure 6a) with the corresponding detrimental effect on the efficiency of energy input under existing experimental conditions.

The XRD patterns in Figure 9 demonstrate that all the samples contain the same phases such as α-W, β-W/W_3_O, and an FCC phase irrespective of the energy input level. Again, the higher energy input resulted in the increased content of the β-W/W_3_O phases. The FCC phase obtained after electric explosion at charging voltages above 22 kV had the crystalline lattice parameter a = 3.61Å and was identified as an α-Cu(Ni) solid solution earlier detected when studying the Cu(Ni) nanoparticles produced via electric explosion [37].

Table 3 demonstrates the the data on measuring the coherent scattering area sizes d_csr_ and crystalline lattice microdistortions Δd/d obtained from the XRD peak broadening of phases detected in W/Cu/Ni-Cr powders. Analogously to the above–described data for W/Cu/Zn (Table 2) both these characteristics reduce as the *E/ΣE_s_* ratio is increased.

The results obtained from the above-described experiments demonstrate that the electric explosion method allows producing micron-and nano-sized powders directly from dissimilar metal wires. It is worth noting that the energy input level had different effects on the particle size of W/Cu/Zn and W/Cu/Ni/Cr powders. For W/Cu/Zn powders the micron particle size distribution became more narrow when increasing the charging voltage *U_0_* (energy input) while coarse >10 μm in size particles retained their presence in W/Cu/Ni/Cr powders. This fact can be explained by specific heating conditions existing on a high electrical-resistance nickel-chromium wire while passing a current pulse through it. The current is limited by high electric resistance of the wire and, therefore, the explosion products are represented by both micron- and nanosized droplets without the gas phase [28]. The crossover to current oscillation regime at *U_0_* > 22 kV interferes with the energy input either into the wire or its explosion products so that the micron-sized Ni-Cr particles are retained.

It can be concluded that alloying copper particles by nickel in W/Cu powders by means of using nickel-chromium wires is almost useless because of obtaining inhomogeneous α-Cu(Ni) particle size distributions. The more promising approach here may be to use W/Cu/Ni wires because of lower electric resistance of Ni or CuNi alloys as compared to that of Ni-Cr.

For the W/Cu-Zn powders, the increase in *U_0_* results in reducing the micron-size particle size distribution range. In our opinion, this is due to lower specific resistance of copper as compared to that of tungsten and, therefore, achieving of a higher energy input ratio *E*/Σ*E_s_*, which, however, does not result in full sublimation of copper as follows from the absence of any Zn-base phases in the powders obtained [31].

The results obtained in the course of this work indicate on the necessity of further investigations to be focused on optimization of elemental composition of the powders. It is known [1] that the majority of W-Cu composites contain not less than 30 wt.% of copper. Our results show that the content of tungsten in our powders is 65–60 wt.%, i.e., less that of the intertwined wires (71–73 wt.%). Enriching the produced powder with tungsten to >70 wt.% may be achieved by using the tungsten wires of large diameters as well as adjusting the separation procedure.

## 4. Conclusions

Simultaneous electric explosion of the intertwined W/Cu-Zn and W/Cu/Ni-Cr wires allowed producing W/Cu/Zn and W/Cu/Ni/Cr powders containing micron- and nanosized particles. As shown, the increase in the energy input into the W/Cu-Zn wire from 0.5E_s_ to 1.25E_s_ resulted in reducing the mean particle size from 3.6 to 2.8 μm. The powder was composed of phases as follows: α-W, β-W/W_3_O, and FCC α-Cu(Zn) with crystalline lattice parameter a ≈ 3.629 A.

Increasing the energy input into W/Cu/79Ni/21Cr wires from 0.5E_s_ to 1.1E_s_ reduced the mean particle size from 4.4 to 3.6 μm. The phases detected in the powder were α-W, β-W/W_3_O and FCC α-Cu(Ni) with crystalline lattice parameter a ≈ 3.61 A. The particle size analysis and EDS elemental distribution indicated the presence of micron-sized Ni-Cr particles. Such a finding demonstrated the inefficiency of using nickel-chromium wires for electric explosion production of copper-nickel-containing powders and necessity of replacing them with nickel ones.

As shown by varying the wire diameter it is possible to obtain homogeneous mixtures of micron- and nanosized particles with various element content ratio values, which allows for synthesizing W-30 wt.%(Cu, Ni, Zn, etc.) powders possessing the desired functional characteristics.

It was shown by the above-discussed results that electric explosion of intertwined dissimilar metal wires is a promising method for synthesizing homogeneous powder mixtures composed of dissimilar metal particles with different particle-size distributions that allow tailoring the characteristics of the powder feedstocks intended for 3D extrusion printing.

## Figures and Tables

**Figure 1 materials-16-00955-f001:**
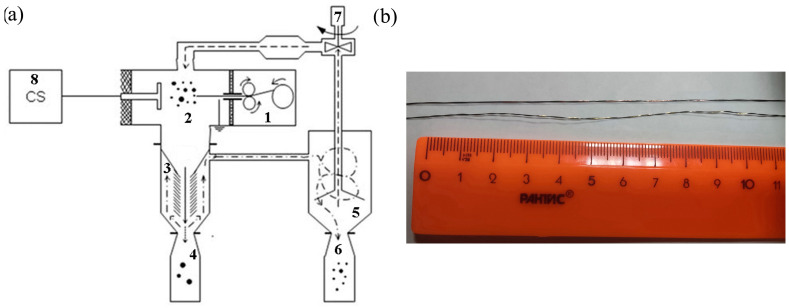
Schematic diagram (**a**) and view of intertwined dissimilar metal wires (**b**).

**Figure 2 materials-16-00955-f002:**
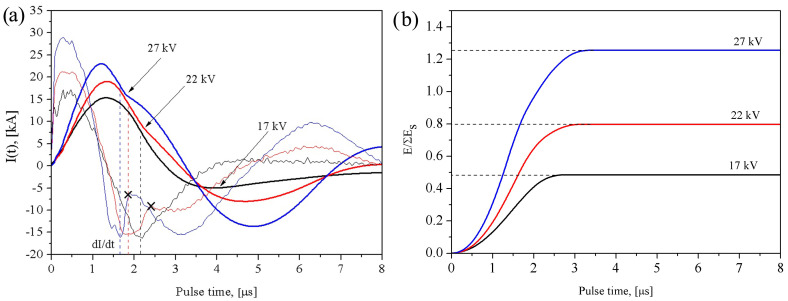
Temporal oscillogram curves of current *I*(*t*) (**a**) and energy *E*/Σ*E_s_* (**b**) for electrical explosion of W/Cu63Zn37(brass) wires. The cross marks denote transition to the arc discharge regime.

**Figure 3 materials-16-00955-f003:**
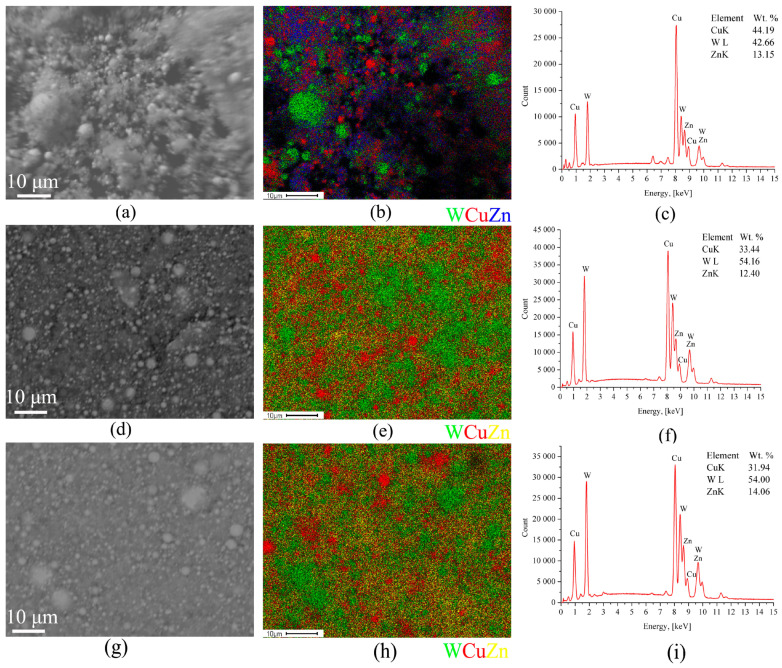
The BSE SEM image of the W/Cu/Zn particles (**a**) and EDS element distribution maps: (**a**–**c**) *E*/Σ*E_s_* = 0.5, (**d**–**f**) *E*/Σ*E_s_* = 0.8, (**g**–**i**) *E*/Σ*E_s_* = 1.25.

**Figure 4 materials-16-00955-f004:**
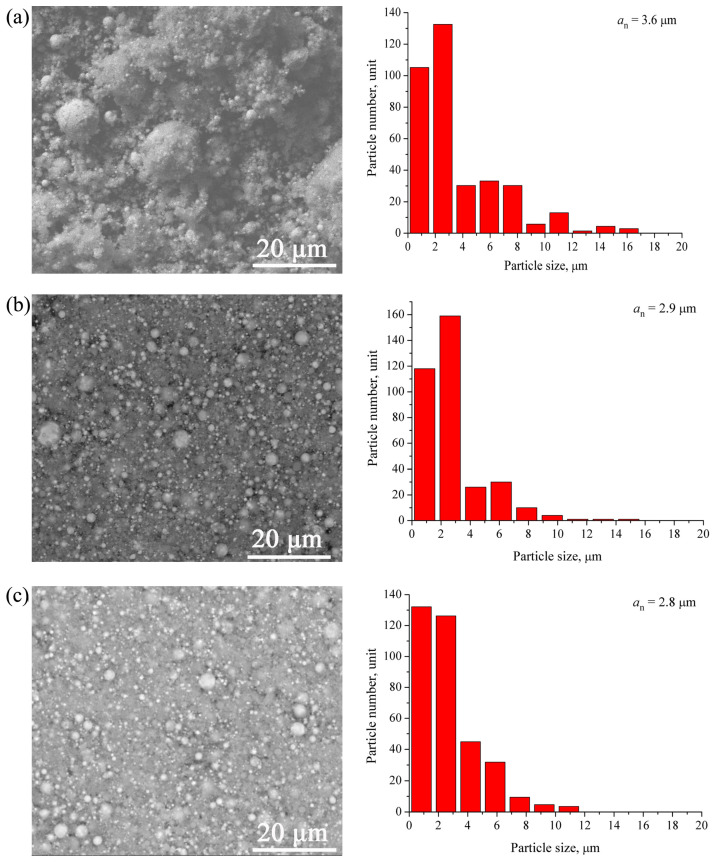
The BSE SEM images and corresponding particle size distributions of W/Cu/Zn powders. (**a**) *E*/Σ*E_s_* = 0.5, (**b**) *E*/Σ*E_s_* = 0.8, (**c**) *E*/Σ*E_s_* = 1.25.

**Figure 5 materials-16-00955-f005:**
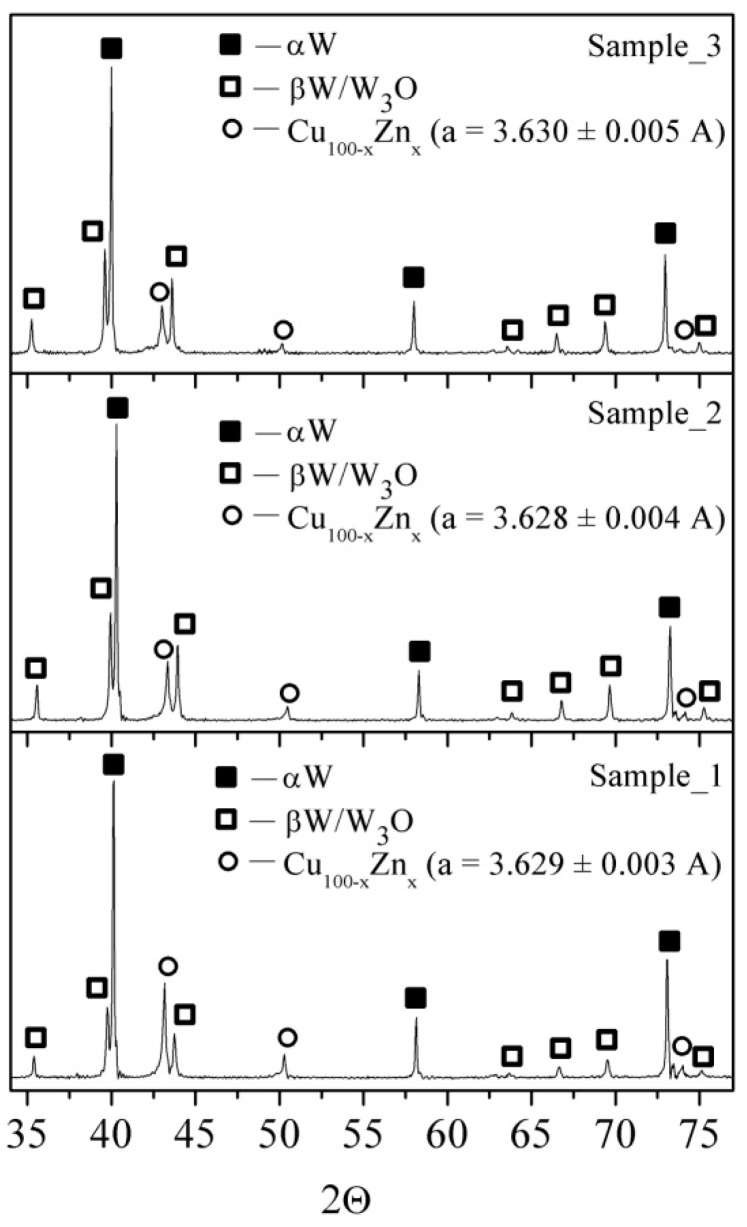
The XRD patterns of the W/Cu/Zn as-exploded powder: Sample_1—*E*/Σ*E_s_* = 0.5, Sample_2—*E*/Σ*E_s_* = 0.8, Sample_3—*E*/Σ*E_s_* = 1.25.

**Figure 6 materials-16-00955-f006:**
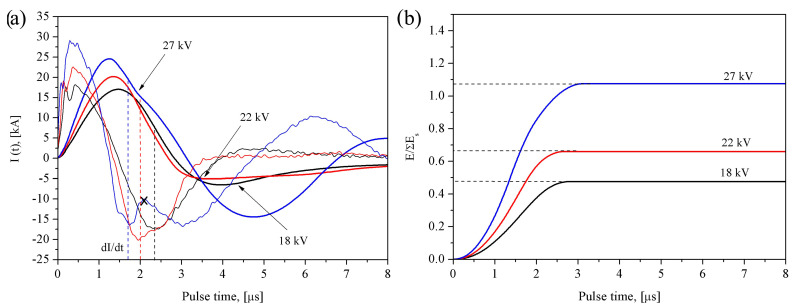
Temporal oscillogram curves of current *I*(*t*) (**a**) and energy *E*/Σ*E_s_* (**b**) for electrical explosion of W/Cu/Ni79Cr21 wires. The cross marks denote transition to the arc discharge regime.

**Figure 7 materials-16-00955-f007:**
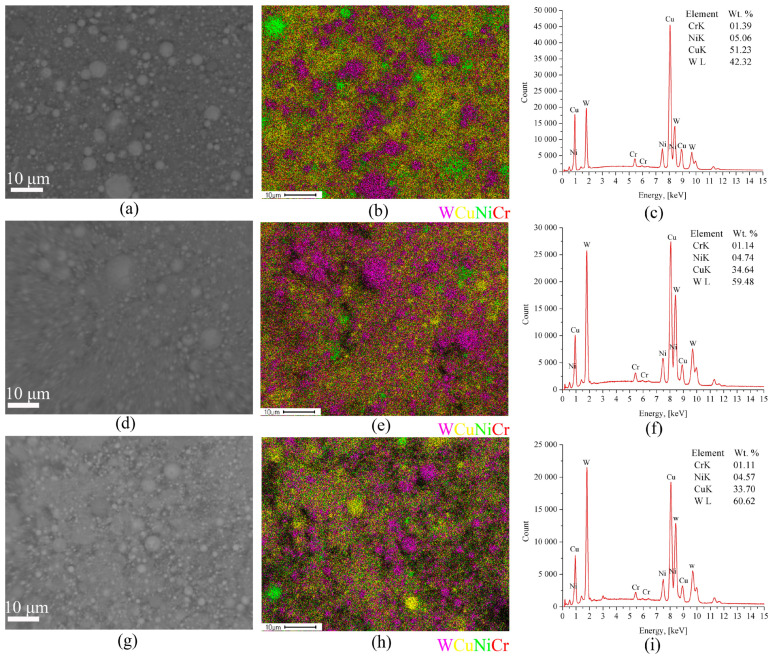
The SEM image of the W/Cu/Ni-Cr particles and EDS element distribution maps: (**a**–**c**) *E*/Σ*E_s_* = 0.5, (**d**–**f**) *E*/Σ*E_s_* = 0.7, (**g**–**i**) *E*/Σ*E_s_* = 1.1.

**Figure 8 materials-16-00955-f008:**
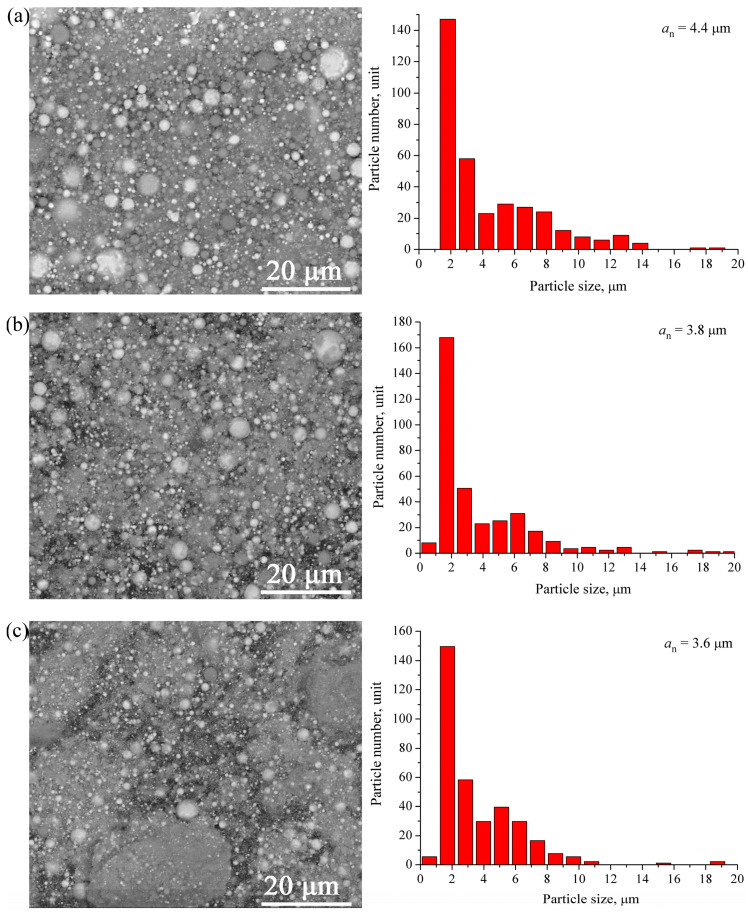
The BSE SEM image and particle size distribution of W/Cu/Ni-Cr. (**a**) *E*/Σ*E_s_* = 0.5, (**b**) *E*/Σ*E_s_* = 0.7, (**c**) *E*/Σ*E_s_* = 1.1.

**Figure 9 materials-16-00955-f009:**
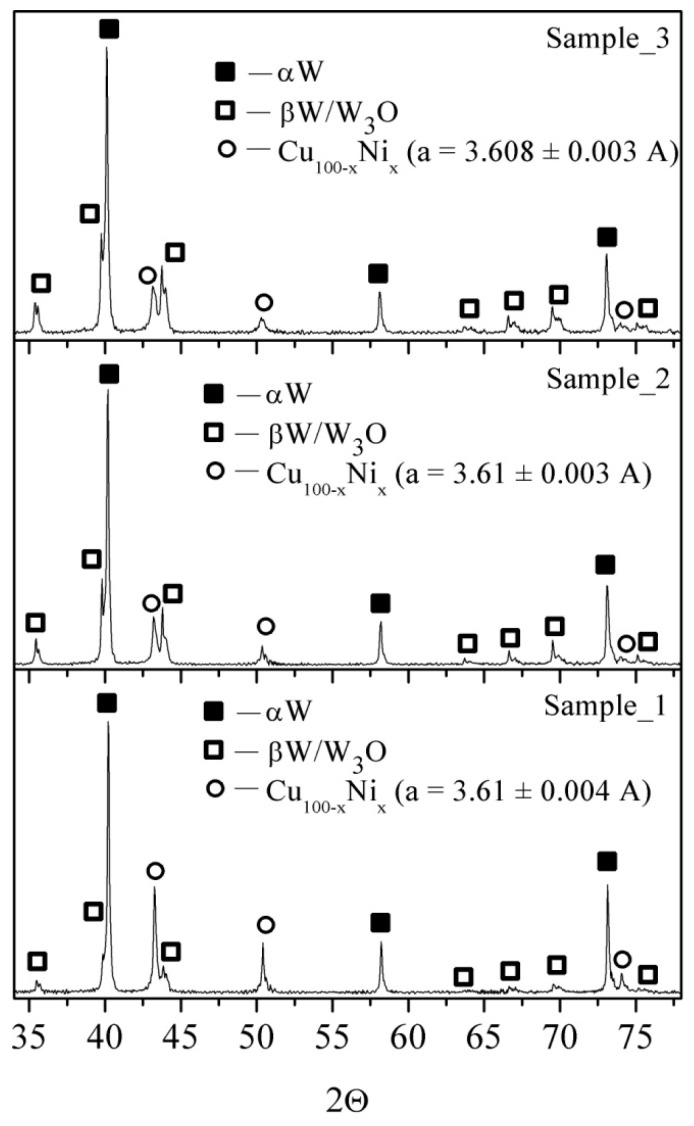
XRD patterns of the W/Cu/Ni/Cr as-exploded powder. Sample_1—*E*/Σ*E_s_* = 0.5, Sample_2—*E*/Σ*E_s_* = 0.7, Sample_3—*E*/Σ*E_s_* = 1.1.

**Table 1 materials-16-00955-t001:** Electric explosion parameters used for producing W/Cu/Zn and W/Cu/Ni/Cr powders.

Composition	Wire Diameter, mm			Wire Length, mm	C, μF	U_0_, kV	ΣE_s_, J	P, MPa
	W	Cu_63_Zn_37_	-					
W/Cu/Zn	0.24	0.22	-	70	1.26	17	261	0.3
						22		
						27		
W/Cu/Ni/Cr	W	Cu	Ni_79_Cr_21_					
	0.24	0.20	0.1	70	1.26	18	310	0.3

**Table 2 materials-16-00955-t002:** Structural characteristics of W/Cu/Zn powders.

Phase	Sample_1		Sample_2		Sample_3	
	d_csr_, nm	Δd/d	d_csr_, nm	Δd/d	d_csr_, nm	Δd/d
α-Cu(Zn)	58 ± 14	0.129 ± 0.018	27 ± 5	0.069 ± 0.024	34 ± 5	0.026 ± 0.020
α-W	174 ± 53	0.083 ± 0.021	48 ± 2	0.012 ± 0.005	46 ± 2	0.012 ± 0.006
β-W/W_3_O	56 ± 8	0.199 ± 0.102	48 ± 4	0.030 ± 0.008	35 ± 6	0.042 ± 0.036

**Table 3 materials-16-00955-t003:** Structural characteristics of W/Cu/Ni/Cr powders.

Phase	Sample_1		Sample_2		Sample_3	
	d_csr_, nm	Δd/d	d_csr_, nm	Δd/d	d_csr_, nm	Δd/d
α-Cu(Ni)	4721	0.089 ± 0.022	22 ± 9	0.084 ± 0.018	20 ± 15	0.055 ± 0.034
α-W	237 ± 22	0.048 ± 0.023	123 ± 31	0.037 ± 0.012	69 ± 12	0.032 ± 0.011
β-W/W_3_O	35 ± 12	0.137 ± 0.072	26 ± 11	0.077 ± 0.038	27 ± 13	0.068 ± 0.041

## Data Availability

The data presented in this study are available on request from the corresponding author.

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
