# Peer review of "Micron- and Nanosized Alloy Particles Made by Electric Explosion of W/Cu-Zn and W/Cu/Ni-Cr Intertwined Wires for 3D Extrusion Feedstock"

_materials, 2023, doi:10.3390/ma16030955_

Round 1
Reviewer 1 Report
The author synthesized the multimetalllic W/Cu-Zn and W/Cu/Ni-Cr micro- and nanoparticles via electrical explosion method. The influences of different energy level on the electrical characteristics and exploded products were investigated. The research is relatively complete; however, there are still some issues.
Q1: It is necessary to show more experimental details, for example, your load structure of the intertwined wires, and your collection method. Are the intertwined wires identical for every explosion? And I wonder if the difference of the intertwined wires will influence the discharge?
Q2: From the current waveforms in Fig 1(a) and Fig 5(a), it seems have no characteristics of multi-wire explosion, it is more like the short circuit discharge mode. So, I think the discharge may mainly happen in W wire, for example, the surface breakdown. In the author’s opinion, if consider that the copper wire participates in the mainly discharge process, more evidences are needed, like the one single wire current in the multi-wire system, or the temporal-spatial resolution images.
Q3: Typically, synthesis materials via electrical explosion method, the products are usually in nano-scale, especially for the overheat coefficient >1. Why micron-particles are dominant in your work? In SEM results, the view is too big. Obviously there exist some cluster/plate-like structures, which may be composed by the nanomaterials. In a word, more microcosmic details are needed to be investigated.
Q4: Researches about intertwined wires explosion to produce multimetalllic alloy powders have been made by former researchers (for example, doi: 10.1016/j.powtec.2022.117491). The results show that one single particle is composed by multi-element; however, in your work, the powders are simple accumulation by different particles with single-element. Can you explain? Further TEM characterization is needed?
Q5: In figure 3(a) and 3(b), the SEM images have obviously differences. Especially in figure 3(a), the products seem like solidified chunks or agglomerates. What makes the differences? Furthermore, I wonder how you make the particle size statistic under such condition? Is it accurate?
Q6: In page 4, line 125 to 131. The author used the EDS results to give the wt.% ratio of Cu and Zn, and the ratio was compared with the ratio of the raw alloy wire. I think such a quantitative analysis is unsuitable. First, EDS characterization is only a semi-quantitative tool. Second, your result is a local view, it can not represent the whole products; furthermore, the measurement depth is also needed to be considered.
Q7: What is the cooling medium adopted in the study? Air or Ar or others? Why adopted 0.3 MPa?
Author Response
Dear Sir!
The authors of the manuscript are very thankful to the reviewers for their valuable, professional and useful comments and questions that allowed improving the understanding and readability of the manuscript. We addressed to each of the comments and provided the responses both below and throughout the manuscript.
My best regards
Prof. Marat Lerner

Reviewer 2 Report
In this manuscript, the authors studied W/Cu/Zn and W/Cu/Ni/Cr based materials as wires. The authors provided several analytical results such as SEM and XRD of the materials and discussed the difference under different energy inputs. In general, the manuscript is well organized. However, the authors failed to give a clear comparison and conclusion of the benefits of the Cu alloy material compared to the original materials. Language improvement is needed for a more accurate description before publication.
P1-P2. L45-46. What is the optimized ratio of alloying Cu with other metal components as referenced?
P2. Experimental Procedure session. What are the procedures for synthesizing the W/Cu-Zn and W/Cu/Ni/Cr material?
P3. L88-90. What is the relation between particle size distribution and energy input here?
P3. L106-107. Provide a brief summary here for the energy input routes reported in reference 29.
P4. Figure 2. Provide scale bar for each of the SEM images. For 33% Cu, 49% W and 18% Zn, it is difficult to observe any of the particles, need to redesign the experiments for high quality images.
P5. L145-152. P6. Figure 4. The XRD spectra of samples 2 and 3 have minimum differences. Is this also related to the different energy inputs? Also, what is the main reason that only β-W/WO3 is sensitive to energy input difference, but α-W is not?
P7. Figure 6. Scale bar should be included in each image. The image of 1.3wt% Cr has low quality and resolution, it is extremely difficult to observe anything in this image.
P8. L175-177. Uncleared description.

Author Response

(The authors gave the same response as above.)

Reviewer 3 Report
The manuscript must be revised carefully to cover the following comments and to improve the content and then the conclusions should be updated after adding the missing figures' description and discussion:
1. Line 83, The particle size distribution was reconstructed from measuring the particle sizes in those SEM micrographs. Add the details of the method used to measure the grain size?
2. Figure 2 (a), how did the authors identify the W and Cu particles in the BSE image (a). Evidence should be added to support this part.
1.1. Add (a), (b), (c), (d),...to identify the images in Figure 2.
1.2. Add the overlay map to display the overall elemental distribution in the selected region.
1.3. Add the EDS spectrum corresponding to the elemental mapping and the obtained quantitative chemical analysis.
2. Figure 3 is not cited in the text, it must be mentioned in the text and described and discussed.
3. Line 145, it is mentioned Table 17. 22 and 27J shows?? Check and revise. If there are missing table, it must be added, cited, described, and discussed in the manuscript.
4. Figure 4 is not cited in the text, it must be mentioned in the text and described and discussed.
5. Figure 4, revise (x or a ) in the CuZn phase, more information must be added in the form of a table to include the peak (phase) reference, the crystal structure, the crystallite size, lattice strain, and any necessary structural information from XRD?
6. Figure 6:
6.1. Add (a), (b),...to identify the images in Figure 6
6.2. Add the overlay map to display the overall elemental distribution in the selected region.
6.3. The EDS spectrum corresponding to the elemental mapping and the obtained quantitative chemical analysis.
7. Apply the comment # 5 on Figure 8.
8. Figure 8 is not cited in the text, it must be mentioned in the text and described and discussed.
9. Line 193, the authors mentioned the above-mentioned experiments…… the figure/table numbers must be mentioned clearly to allow the readers to follow the content and link the presented figures/tables with the related discussion?
Author Response

(The authors gave the same response as above.)

Round 2
Reviewer 2 Report
The authors have addressed all the questions and re-edited the manuscript, it is in good shape for publication.
Reviewer 3 Report
Thanks to the authors for considering all comments and suggestions.